# Anomalous coupling studies with intact protons at the LHC

Christophe Royon[1]

**1** The University of Kansas, Lawrence, USA
* christophe.royon@ku.edu

October 14, 2022

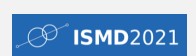

*51st International Symposium on Multiparticle Dynamics (ISMD2022)*
*Pitlochry, Scottish Highlands, 1-5 August 2022*
doi:[10.21468/SciPostPhysProc.?](https://doi.org/10.21468/SciPostPhysProc.?)

## Abstract

**We describe the reaches on quartic $\gamma\gamma\gamma\gamma$, $\gamma\gamma WW$, $\gamma\gamma ZZ$, $\gamma\gamma\gamma Z$, $\gamma\gamma t\bar{t}$ anomalous couplings at the LHC using intact protons in the final state measured in AFP in ATLAS or PPS in CMS-TOTEM.**

## 1 The LHC as a $\gamma\gamma$ collider

We are interested in the exclusive production of diphoton, $WW$, $ZZ$ bosons, $Z\gamma$ and $t\bar{t}$, as shown in Fig. 1, left in the case of diphoton production, where the photons and the decay products of the $W$, $Z$ bosons and the top quarks are measured in the main ATLAS or CMS detector and the intact protons in dedicated roman pot detectors located at about 220 m from the interaction point. The exclusive production can be due to an exchange of pair of gluons (in order to get a colorless object) or photons. At high diffractive masses of about 450 GeV, the exchange of photons dominate by more than two orders of magnitude [1–4]. We can thus consider the LHC as a $\gamma\gamma$ collider. The

acceptance of the forward detectors is shown in Fig. 1, right, for standard high luminosity runs at low $\beta^*$ at the LHC when one approaches the beam at 15 or $20\sigma$. We see that the diffractive mass acceptance starts at about 450 GeV up to about 2 TeV. Special runs at higher $\beta^*$ allow accessing lower diffractive mass acceptances. The CMS and TOTEM collaborations collected about 115 fb$^{-1}$ of data in 2016-2018 at low $\beta^*$.

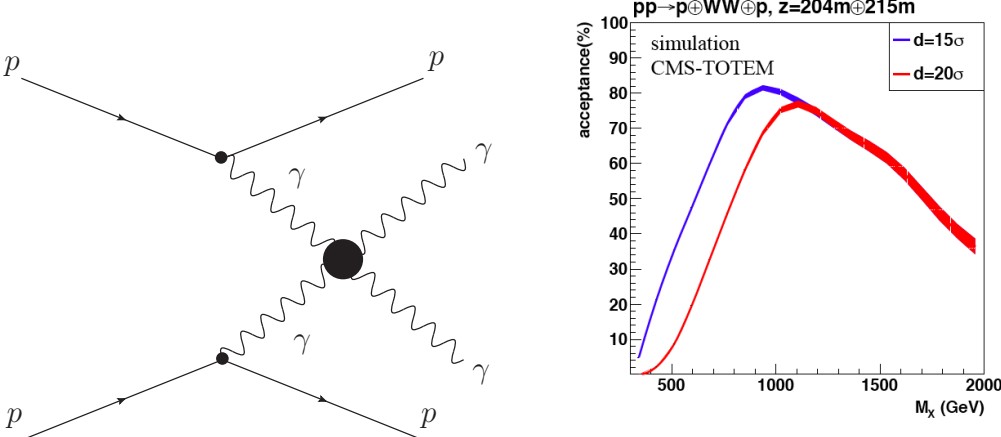

Figure 1: Left: Exclusive diphoton production via photon exchanges. Right: Diffractive mass acceptance for two distances from the beam at 15 and $20\sigma$.

The photon-induced production of $\gamma\gamma$, $WW$, $ZZ$, $Z\gamma$ and $t\bar{t}$ with intact protons lead to very clean events where all particles produced in the final state are detected and measured like at LEP. The main background to the exclusive production is due to pile up events where $\gamma\gamma$, $WW$, $ZZ$, $Z\gamma$ and $t\bar{t}$ are produced non-exclusively and the protons are destroyed. Intact protons are produced from additional interactions called pile up. Kinematic conservation such as the equality of the proton and $\gamma\gamma$ system missing mass and rapidity for signal events allow rejecting most of the pile up events as shown in Fig 2.

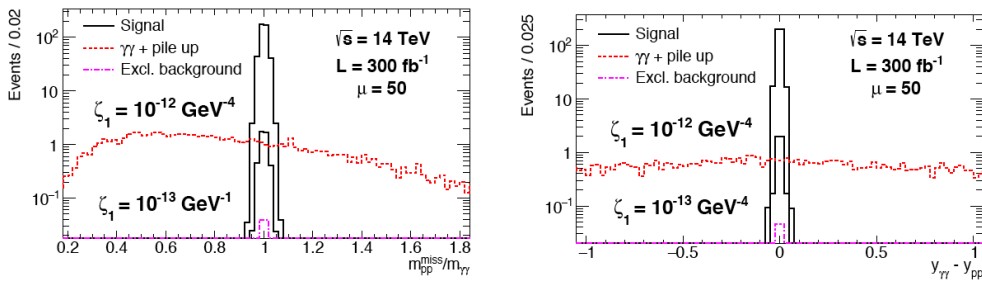

Figure 2: Ratio of the diphoton mass and the proton missing mass (left) and difference in rapidity between the diphoton and the two proton system between signal and pile up events.

The motivation to look for the exclusive production of diphotons as an example is to look for quartic $\gamma\gamma\gamma\gamma$ anomalous couplings that could be a sign of new physics. We can define two effective

operators at low energy in the Lagrangian

$$L_{4\gamma} = \zeta_1^\gamma F_{\mu\nu} F^{\mu\nu} F_{\rho\sigma} F^{\rho\sigma} + \zeta_2^\gamma F_{\mu\nu} F^{\nu\rho} F_{\rho\lambda} F^{\lambda\mu}. \tag{1}$$

This $\gamma\gamma\gamma\gamma$ coupling can be modified in a model independent way by loops of heavy charged particles

$$\zeta_1 = \alpha_{em}^2 Q^4 m^{-4} N c_{1,s} \tag{2}$$

where the coupling depends only on $Q^4 m^{-4}$, $Q$ and $m$ being the charge and mass of the charged particle and on spin, $c_{1,s}$. This can lead to $\zeta_1$ of the order of $10^{-14}$-$10^{-13}$ GeV$^{-4}$ depending on the models. $\zeta_1$ can also be modified by neutral particles at tree level (extensions of the SM including scalar, pseudo-scalar, and spin-2 resonances that couple to the photon)

$$\zeta_1 = (f_s m)^{-2} d_{1,s} \tag{3}$$

where $f_s$ is the $\gamma\gamma X$ coupling of the new particle $X$ to the photon, and $d_{1,s}$ depends on the spin of the particle. For instance, 2 TeV dilatons lead to $\zeta_1 \sim 10^{-13}$ GeV$^{-4}$. This coupling can also be modified by the existence of axion-like particles (ALP).

## 2 Sensitivity to quartic $\gamma\gamma\gamma\gamma$ anomalous couplings and to the production of axion-like particles

Using the matching between the diphoton and the proton missing mass and rapidity distributions as described in the previous section, as well as requiring diphoton produced at high mass and back-to-back, a negligible background is found for a luminosity of 300 fb$^{-1}$ that leads to a sensitivity up to a few $10^{-15}$ GeV$^{-4}$ on $\zeta_1$, better by 2 orders of magnitude with respect to "standard" methods at the LHC. Exclusivity cuts using proton tagging are crucial to suppress the pile up background which is 80.2 events for 300 fb$^{-1}$) before matching.

This result can be directly applied to the production of ALPs that can be produced as a resonance decaying into two photons or as a loop coupled to photons. The reach in the coupling versus mass of the axion-like particles is shown in Fig. 3 in pp collisions at the LHC with 300 fb$^{-1}$ [5]. We gain about two orders of magnitude in the sensitivity to ALPs with respect to standard methods at the LHC and we even reach a domain at high ALP mass at the LHC that was not covered before. In addition, as shown in Fig. 3, the production of ALPs via photon exchanges in heavy ion runs ($pPb$, $PbPb$ and $ArAr$ collisions) allows covering the intermediate domain in ALP masses since the cross section is increased by a factor $Z^4$ [6].

## 3 Sensitivity to $\gamma\gamma WW$, $\gamma\gamma ZZ$, $\gamma\gamma\gamma Z$, $\gamma\gamma t\bar{t}$ anomalous couplings

The search for anomalous $\gamma\gamma WW$ and $\gamma\gamma ZZ$ anomalous couplings can be performed using the hadronic decays of the $W$ and $Z$ bosons [7–9]. In Fig. 4, left, is displayed the $WW$ mass distribution $m_{ww}$ as measured using the roman pot detectors for exclusive $WW$ production for SM and two values of anomalous $\gamma\gamma WW$ couplings. We see that anomalous coupling events have a tendency to be produced at higher mass with respect to the SM production. Looking for anomalous $WW$ production will thus benefit from the hadronic decay of the $W$ bosons (since the jet background

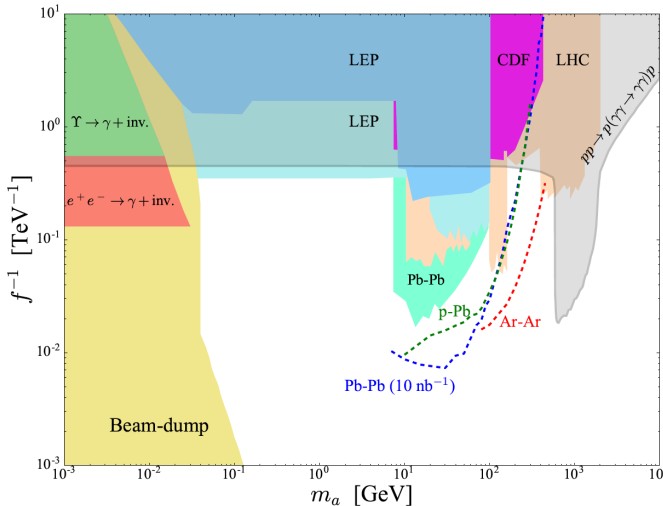

Figure 3: Sensitivity to the production of axion-like particles in the coupling versus mass plane for $pp$, $pPb$, $PbPb$ and $ArAr$ interactions at the LHC.

is lower at higher masses) whereas the SM production can be measured using the leptonic decays of the $W$ bosons where the background is smaller.

For an anomalous coupling of $10^{-6}$ GeV$^{-2}$, we expect about 110 events for a background of 87 due to pile up events in the hadronic decay channel of the $W$ bosons for 300 fb$^{-1}$ [9]. The sensitivity is up to $a_0 = 3.7 \ 10^{-7}$ GeV$^{-2}$ (the present limits using exclusive production of $WW$ at medium luminosity with low pile up and without proton tagging led to limits of $\sim 10^{-4}$ GeV$^{-2}$). The sensitivity contour plot in the anomalous coupling $a_C^W$ versus $a_0^W$ plane with 300 fb$^{-1}$ of data at the LHC is shown in Fig. 4, right.

It is also possible to observe the SM $WW$ exclusive production in the leptonic decay channel of the $W$ bosons with 300 fb$^{-1}$. After selection, we predict about 50 events to be measured with 2 background events [9], which can lead to the first possible observation of exclusive $WW$ production at high $WW$ mass.

The search for $\gamma\gamma\gamma Z$ anomalous coupling at the LHC can also be performed when the $Z$ boson decays leptonically or hadronically [10]. It leads to the best expected reach at the LHC by about three orders of magnitude compared to the standard search performed in looking for the $Z$ boson decay into three photons that is challenging in a high pile up environment.

The search for $\gamma\gamma t\bar{t}$ anomalous coupling can be performed in the leptonic and semi-leptonic decays of $t$ and $\bar{t}$ in order to avoid the large background of the pure hadronic decays, due to the standard non exclusive $t\bar{t}$ production and intact protons from pile up. After requesting a high $t\bar{t}$ mass measured using the proton roman pot detectors and requesting a matching between the $pp$ and $t\bar{t}$ measurements, we obtain the results described in Table 1. The matching between the protons and $t\bar{t}$ mass and rapidity information does not reject fully the pile up background because of the presence of the neutrino originating from the top quark decay in the semi-leptonic mode and the worse resolution on the mass measured in the CMS and ATLAS main detectors. The reach on $\gamma\gamma t\bar{t}$ anomalous coupling benefits strongly from the resolution of the timing detectors that allow to measure the proton time-of-flight and to constrain the protons to originate from the same interaction vertex as the $t\bar{t}$ as illustrated in Table 1.

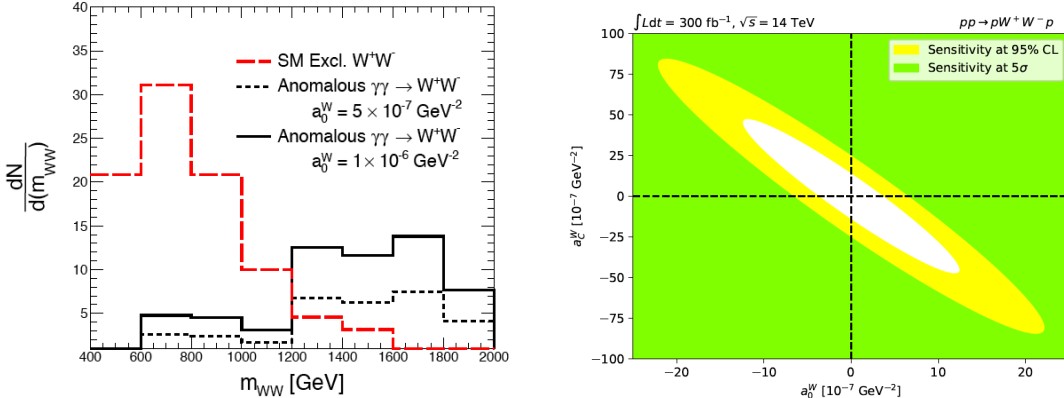

Figure 4: Left: $WW$ mass distribution $m_{ww}$ as measured using the roman pot detectors for exclusive $WW$ production for SM and two values of anomalous $\gamma\gamma WW$ couplings. Right: Sensitivity contour plot in the anomalous coupling $a_C^W$ versus $a_0^W$ plane with 300 fb$^{-1}$ of data at the LHC.

| Coupling [$10^{-11}$ GeV$^{-4}$] | 95% CL | $5\sigma$ | 95% CL (60 ps) | $5\sigma$ (60 ps) | 95% CL (20 ps) | $5\sigma$ (20 ps) |
|---|---|---|---|---|---|---|
| $\zeta_1$ | 1.5 | 2.5 | 1.1 | 1.9 | 0.74 | 1.5 |
| $\zeta_2$ | 1.4 | 2.4 | 1.0 | 1.7 | 0.70 | 1.4 |
| $\zeta_3$ | 1.4 | 2.4 | 1.0 | 1.7 | 0.70 | 1.4 |
| $\zeta_4$ | 1.5 | 2.5 | 1.0 | 1.8 | 0.73 | 1.4 |
| $\zeta_5$ | 1.2 | 2.0 | 0.84 | 1.5 | 0.60 | 1.2 |
| $\zeta_6$ | 1.3 | 2.2 | 0.92 | 1.6 | 0.66 | 1.3 |

Table 1: 95% CL and $5\sigma$ projected limits on each of the couplings, setting the other ones to zero. Multiple timing detector performance scenarios are considered: no timing information, timing detector resolution $\sigma_t = 60$ ps, and timing detector resolution $\sigma_t = 20$ ps.

## 4  Conclusion

In this short report, we presented the prospects on quartic $\gamma\gamma\gamma\gamma$, $\gamma\gamma WW$, $\gamma\gamma ZZ$, $\gamma\gamma\gamma Z$, $\gamma\gamma t\bar{t}$ anomalous coupling using proton tagging at the LHC. The exclusive production at high diffractive masses is dominated by photon exchanges at the LHC that can be considered as a $\gamma\gamma$ collider. It leads to clean events like at LEP where all particles are measured in the final state and thus to sensitivities increased by two or three orders of magnitude on quartic anomalous coupling and axion-like particle production with respect to more standard methods at the LHC.

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
