# Peer review of "Anomalous coupling studies with intact protons at the LHC"

_SciPost Physics Proceedings_

## Round 1 · Referee Report · Anonymous (Referee 1) · 2022-12-14

Report
This contribution presents a range of promising studies relating to anomalous couplings with intact protons at the LHC. I am happy to recommend publication after the relatively minor comments are accounted for below.
1) Just below Fig. 1. 'can be detected' is perhaps more precise than 'are detected'.
2) Same paragraph as above. The definition of pile up could be a bit more precise.
3) Above (2) I would not describe this as 'model independent' as it explicitly assumes new heavy charged particles, which is certainly a model. So perhaps 'general way'.
4) The comment in the second paragraph of Section 2 is definitely too strong. There are inclusive constraints on ALPs due to inclusive diphoton production as described in e.g. 2102.08971, see Fig. 4 (right), where one can see that the constraints in this region are already comparable. So this needs to be very much softened.

---

## Editorial Decision

unknown